# Novel Insights into Pigment Composition and Molecular Mechanisms Governing Flower Coloration in Rose Cultivars Exhibiting Diverse Petal Hues

**DOI:** 10.3390/plants13233353

**Published:** 2024-11-29

**Authors:** Yingxia Cheng, Yanling Tian, Pengyu Guo, Junjie Luo, Chan Xu, Yang Zhang, Guoping Chen, Qiaoli Xie, Zongli Hu

**Affiliations:** 1Bioengineering College, Chongqing University, Chongqing 400044, China; 18397910815@163.com (Y.C.); tianna0816@163.com (Y.T.); guopengyucqu@163.com (P.G.); 15310835647@163.com (J.L.); chenguoping@cqu.edu.cn (G.C.); 2Key Laboratory of Biorheological Science and Technology, Chongqing University, Ministry of Education, Chongqing 400044, China; 3Chongqing Academy of Agricultural Sciences, Agricultural Science Avenue, Chongqing 400039, China; xchan8808@163.com; 4College of Life Sciences, Sichuan University, Chengdu 610065, China; yang.zhang@scu.edu.cn

**Keywords:** rose (*Rosa hybrid*), anthocyanin biosynthesis, carotenoid biosynthesis, metabolome, transcriptome, transcription factors

## Abstract

The pigmentation of various components leads to different colors of roses. However, the intricate molecular machinery and metabolic pathways underlying rose pigmentation remain largely unexplored. In this study, we determined that pink and black-red petals contain abundant anthocyanins, reaching concentrations of 800 μg/g and 1400 μg/g, respectively, significantly surpassing those in white and yellow petals. We identified 22 key anthocyanin components, predominantly cyanidin, pelargonidin, delphinidin, peonidin, and petunidin, which were preferentially enriched in pink and black-red petals. Additionally, we confirmed the presence of five carotenoid species—lutein, zeaxanthin, ζ-carotene, α-carotene, and β-carotene—with zeaxanthin and carotenoids notably accumulating in yellow petals at significantly higher levels compared with other colors. Furthermore, RNA-seq and qRT-PCR analyses revealed the association between pigment accumulation and the expression patterns of genes involved in anthocyanin and carotenoid biosynthesis pathways. Through promoter core element prediction and transcriptional metabolic co-expression analyses, we found that the MYB transcription factor likely positively modulates the expressions of key biosynthetic genes such as CHS, F3′H, and DFR, while the NAC transcription factor enhances the transcriptional activities of PSY, ZISO, and LYCB. Overall, this study explores the components of flower color, unravels the synthesis of anthocyanins and carotenoids, identifies regulatory factors, and highlights the prospects of rose breeding.

## 1. Introduction

Flower color stands as a pivotal attribute of ornamental plants and a distinctive trait in higher plants. Flower color plays a crucial role in attracting pollinators, thereby facilitating plant reproduction [1,2]. The development of flower color is governed by plant pigments, with the accumulation of flavonoids, carotenoids, and betaines being primarily responsible for petal coloration. Flavonoids, exemplified by anthocyanins and other substances, constitute a significant group of these pigments. A branched pathway in flavonoid synthesis leads to the production of both colorless molecules and colored pigments [3]. Anthocyanins impart hues to flowers ranging from light pink to purple, while carotenoids predominantly yield yellow colors [4]. Notably, anthocyanin is a ubiquitous natural pigment found in vascular plants [5].

The biosynthetic process of anthocyanins is relatively conserved and has been extensively studied across various plant species. This process initiates with phenylalanine, which undergoes a series of transformations catalyzed by several enzymes [6]. Early in biosynthesis, phenylalanine is converted into cinnamic acid by phenylalanine ammonia-lyase (PAL). Subsequent key enzymes involved in flavonoid metabolism include cinnamate-4-hydroxylase (C4H), chalcone synthase (CHS), 4-coumarate-CoA ligase (4CL), flavonoid-3′-hydroxylase (F3′H), flavanone 3-hydroxylase (F3H), and chalcone isomerase (CHI). Dihydroflavonol 4-reductase (DFR), anthocyanidin synthase (ANS), and UDP-flavonoid glucosyltransferase (UFGT) primarily function in later stages. These enzymes are responsible for producing proanthocyanidins (PAs) and anthocyanidins [7,8].

Transcription of the genes encoding these synthases is tightly regulated by the evolutionarily conserved ‘MBW’ complex, comprising R2R3-MYB proteins, basic helix–loop–helix (bHLH) proteins, and WD40 repeat (WDR) proteins [9]. MYB transcription factors (TFs) are pivotal for anthocyanin synthesis, with numerous R2R3-MYB TFs isolated [10]. These TFs primarily enhance anthocyanin biosynthesis, though some exhibit inhibitory effects. Specifically, the R2R3-MYB TF, a key regulator of anthocyanin biosynthesis, directs downstream gene expression, resulting in tissue-specific anthocyanin accumulation [11,12].

Carotenoids are essential secondary metabolites that give plants yellow to red hues and are synthesized and stored in photosynthetic organisms [13]. Beyond their color-imparting role, carotenoids serve as developmental signals and antifungal agents, aiding plant growth and floral scent production. Carotenoid metabolism genes have been extensively characterized in diverse plants through rigorous biochemical and molecular biology research. The key enzymes in carotenoid metabolism include phytoene desaturase (PDS), ζ-carotene desaturase (ZDS), phytoene synthase (PSY), and carotenoid isomerase (CRTISO), which govern the synthesis of straight-chain carotenoids such as ζ-carotene and lycopene. Lycopene, a critical branch point, can be converted into α-carotenoids or β-carotenoids by lycopene β-cyclase (LCYB) and lycopene ε-cyclase (LCYE), respectively. These compounds undergo further hydroxylation and epoxidation reactions, transforming into various luteins, thereby enriching carotenoid diversity [14,15].

Roses are widely used as ornamental plants in gardens, as cut flowers, and for refining essential oils in perfumery and cosmetics [16]. Significant emphasis is placed on rose color as a vital element to capture consumers’ attention. Prior research has shown that rose color formation is regulated by two pigments: anthocyanins dominate the creation of orange, pink, red, and purple hues, while carotenoids contribute to the yellow color [17]. However, the molecular mechanisms and metabolic pathways underlying rose pigmentation remain elusive. In this study, we investigated the inheritance pattern of floral pigmentation in four representative rose varieties: white rose (Bai Xueshan), yellow rose (Jin Xiangyu), pink rose (Litchi), and black-red rose (Black Magic). High-performance liquid chromatography-mass spectrometry (HPLC-MS) and liquid chromatography-mass spectrometry (LC-MS) systems were employed to characterize the anthocyanin and carotenoid compositions in these roses. A comprehensive analysis of metabolites and transcriptomics provided insights into the differential metabolites and molecular mechanisms driving flower color variations. These findings contribute to our understanding of color differentiation in roses and aid in identifying candidate genes for breeding more vibrant and colorful flowers.

## 2. Results

### 2.1. Phenotypic Characteristics and Pigment Composition of Representative Varieties of Roses

The co-accumulation of flavonoids and carotenoids determines the color of flowers. To investigate the biosynthesis of rose pigments, we selected four representative rose varieties with distinct colors: Bai Xueshan (white, W), Jin Xiangyu (yellow, Y), Litchi (pink, P), and Black Magic (black-red, B) (Figure 1a). Upon inspecting these rose varieties, we found that their flower colors matched perfectly with the selected categories. To identify the types of pigments present in the different rose varieties, we analyzed the anthocyanin and carotenoid content in the petals of each. Our results indicated that the petals of the white (W) and yellow (Y) rose varieties contained little to no anthocyanins (Figure 1b). In contrast, the pink (P) and black-red (B) rose varieties exhibited higher anthocyanin contents of 800 µg/g and 1400 µg/g, respectively. Regarding carotenoid content, the yellow (Y) rose variety had the highest concentration of carotenoids at 350 µg/g. The pink (P) rose variety contained a minimal amount of carotenoids, while the white (W) and black-red (B) rose varieties had negligible amounts of carotenoids (Figure 1c).

### 2.2. Differential Accumulation Metabolite (DAM) Analysis of Four Rose Varieties

To gain deeper insights into the chemical mechanisms underlying petal coloration in four rose varieties, we conducted a metabolomics analysis using UPLC-MS/MS (Appendix A). By referencing the local metabolite database, we identified 155, 214, and 282 differential metabolites in comparisons of the Y/W, P/W, and B/W varieties, respectively. Notably, the accumulation of these differential metabolites increased with the darkening of petal color. These metabolites primarily belonged to flavonoids, alkaloids, phenolic acids, tannins, terpenoids, lignans, coumarins, and other categories (Figure 2a and Appendix A). Using the metabolite quantification data, we performed a principal component analysis (PCA) and generated a hierarchical clustering heatmap. The results revealed a strong correlation between biological replicates, suggesting high-quality metabolome quantification (Figure 2b). Furthermore, a clear separation was observed between the petal samples of the W, Y, P, and B varieties, indicating the presence of distinct compounds in each sample (Appendix A). Based on the differential metabolite results, we conducted KEGG pathway enrichment analyses, comparing the Y, P, and B varieties with the W variety to identify the most highly enriched pathway categories. As illustrated in the Appendix A, flavonoid and flavonol biosynthesis, flavonoid biosynthesis, and anthocyanin biosynthesis emerged as the most representative metabolic pathways in all three comparisons, with greater enrichment observed in the P/W and B/W groups. Therefore, we further analyzed the flavonoid DAMs in the P/W and B/W groups.

Our metabolome data analysis identified 65, 114, and 142 types of differentially accumulated flavonoids (DAFs) in the Y/W, P/W, and B/W groups, respectively. Compared with the W variety, 48, 57, and 92 types of DAFs were upregulated in the Y, P, and B varieties, respectively (Appendix A). A further comparison of DAFs across the four samples revealed significant differences in the accumulation of 30 metabolites, primarily flavanols, flavonoids, dihydroflavonols, dihydroflavones, and anthocyanins (Figure 2c and Appendix A). Notably, luteolin and pinocembrin were abundant in the Y samples, while kaempferol was more prevalent in the Y samples compared with the other three groups. The pelargonidin content was higher in the P samples, and most DAFs were heavily accumulated in the B samples, including quercetin, naringenin, dihydroquercetin (taxifolin), and pelargonidin (Figure 2c). These findings suggest that luteolin, pinocembrin, and kaempferol may contribute to the formation of yellow color, while pelargonidin and quercetin are key metabolites in the formation of pink and red colors.

To gain a deeper understanding of anthocyanin accumulation, we further analyzed the relative content of anthocyanins in these samples to identify differentially accumulated anthocyanins (DAAs) (Appendix A). The results showed that more DAAs were detected in the P and B varieties, with 16 and 22 anthocyanins upregulated, respectively. These may be key metabolites affecting rose petal coloration (Appendix A). Since the anthocyanin biosynthesis pathway was found to be significantly enriched in the P/W and B/W groups by KEGG pathway enrichment (Appendix A), we compared the P and B varieties with the W variety to further analyze the DAAs among these three varieties (Figure 2d and Appendix A). The results identified five major DAAs: cyanidin, pelargonidin, delphinidin, peonidin, and petunidin. These DAAs were significantly more abundant in the P and B varieties than in the W variety, with the highest accumulation observed in B. Among them, pelargonidin-3,5-O-diglucoside and pelargonidin-3-O-glucoside were most abundant in the P variety, indicating their importance in pink color formation. In contrast, cyanidin-3-O-(6″-O-p-coumaroyl-2″-O-xylosyl) glucoside and cyanidin-3-O-(6″-O-p-coumaroyl) glucoside were higher in the W variety. The remaining DAA compounds were abundant in the petals of the B variety, suggesting their significant contribution to the development of the deep red color. Thus, the high accumulation of these DAAs in the P and B varieties may have contributed to their color-related changes.

To further investigate the carotenoid composition and content in the petal extracts of the four rose varieties (Bai Xueshan (W), Jin Xiangyu (Y), Litchi (P), and Black Magic (B)), we conducted HPLC and LC-MS analyses. The carotenoid isolates from the W, Y, P, and B samples exhibited spectra consistent with standards, enabling the identification of five major components: lutein, zeaxanthin, ζ-carotene, α-carotene, and β-carotene (Figure 2e–i and Appendix A). The petals of the W and B varieties primarily contained lutein, while all substances were present in the petals of the Y variety. Lutein, zeaxanthin, and ζ-carotene were predominantly found in the petals of the P variety. In summary, our analysis showed that the content of carotenoids, which are the source of yellow color, was significantly higher in the Y variety than in the other tested varieties.

### 2.3. Transcriptome Analysis of Petals of Four Rose Varieties

Utilizing RNA-seq data, we conducted a genome-wide analysis of gene expression and alterations in the expression profiles of four petal samples differing in color. Three biological replicates of 12 petal samples generated a total of 623,488,310 clean reads, representing 93.53 Gb of clean data. Each sample contributed approximately 6 Gb of data, and the Q30 percentage was over 88%, indicating high-quality sequencing results (Appendix A). Thereafter, differential expression analysis was performed using DESeq2 to evaluate the transcriptional profiles of three biological replicas from the W, Y, P, and B samples. This analysis identified a total of 11,316 genes with significantly altered expression levels (DESeq2 FDR < 0.05 | log2(fold change) | > 1). Subsequently, we compared the DEGs across the four samples. Specifically, the counts of DEGs were 6240 for the Y/W group, 5543 for the P/W group, and 5783 for the B/W group. Figure 3a illustrates the distribution of differentially expressed genes (DEGs) across the three comparison groups: Y vs. W, P vs. W, and B vs. W. Hierarchical clustering of significant gene expression changes based on FPKM values revealed distinct transcriptome profiles among the four rose petal colors (Figure 3b). Functional classification and pathway assignments of annotated sequences were conducted using the KEGG database, encompassing systematic analyses of intracellular metabolic pathways and gene functions.

### 2.4. Expression Analysis of Genes Related to Anthocyanin Biosynthesis Pathway

To uncover genes associated with anthocyanin biosynthesis in the RNA-seq dataset, we analyzed the expression patterns of structural genes through heat map visualization, utilizing their FPKM values across petals varying in color (Figure 4a). The findings indicated that a large proportion of genes in the other three color varieties showed higher expression levels compared with the W petal varieties (Figure 4a). PAL, C4H, 4CL, and F3H were more highly expressed in the W sample compared with the other three varieties in the rose samples. Structural genes were predominantly expressed in the Y, P, and B samples, peaking in the B sample and lower in the Y sample (Figure 4a). The differences in gene expression indicate functional variations. To confirm the RNA-seq findings, we quantified flavonoid biosynthesis genes via qRT-PCR (Figure 4b). The results concurred with the RNA-seq data (Figure 4b), thereby validating the reliability of our RNA-seq approach.

### 2.5. Expression Analysis of Genes Related to Anthocyanin

A comparative analysis of carotenoid biosynthesis pathway gene expression in Figure 5 revealed elevated PSY expression in the Y and B varieties compared with the W and P varieties. CHYB and VDE exhibited similar expression patterns, both exhibiting high expression in the B varieties, whereas LYCB displayed the highest expression levels in the Y varieties (Figure 5a). These structural gene expressions correlate with carotenoid composition. Yet other carotenoid genes, such as PDS, ZISO, ZDS, and NCED, showed higher expression in P varieties (Figure 5a). An analysis of carotenoid gene expression revealed that PSY and ZDS positively correlate with carotenoid accumulation (Figure 5a,b). Variations in structural gene expression modulate carotenoid biosynthesis and accumulation, resulting in diverse colors.

### 2.6. Analysis of Transcription Factors Related to the Regulation of Rose Flower Color

MYB, bHLH, WRKY, and NAC TFs regulate key enzyme genes during anthocyanin biosynthesis, influencing color formation. To elucidate petal gene expression and identify TFs linked to color variation, we normalized FPKMs, conducted K-means clustering, and defined seven sub-classes (Appendix A). The genes of Sub-classes 5, 3, and 4 exhibited the highest expression in the Y, P, and B samples, respectively, suggesting their positive association with anthocyanin synthesis and accumulation. In contrast, Sub-class 7 genes were predominantly expressed in the W samples, hinting at a negative correlation with pigment accumulation. Based on these findings, we postulate that the genes of Sub-classes 5, 3, 4, and 7 likely contribute to anthocyanin synthesis and accumulation.

Analyzing TF-related genes in these sub-classes revealed 70 differentially expressed TFs: 17 bHLH, 19 MYB, 23 NAC, and 11 WRKY. Notably, NACs were predominantly expressed in the Y samples, with lower levels in the P and B samples (Figure 6a). Three bHLH (LOC112175393, LOC112199892, and LOC112169727), two MYB (LOC112168782 and LOC112185002), and one WRKY (LOC112176545) were present in Sub-class 3 and were more highly expressed in the P sample. Five bHLH (LOC112188917, LOC112179563, LOC112174687, LOC112185658, and LOC112180575), five MYB (LOC112171864, LOC112194873, LOC112178787, LOC112176685, and LOC112185427), and three WRKY (LOC112192514, LOC112177149, and LOC112164583) were present in Sub-class 4 and highly expressed in the B samples (Appendix A). These findings indicate that these transcription factors likely promote the development of red color in roses. In addition, we examined the expressions of RhMYB1 (LOC112171864), RhMYB113 (LOC112194873), and RhMYB1R1 (LOC112178787) in the petals of four varieties of roses via qRT-PCR (Figure 6b), and both RhMYB1 and RhMYB113 were highly expressed in the P and B samples.

We scrutinized the promoter sequence—spanning 3000 bp upstream of ATG—of a key anthocyanin synthesis gene that showed robust expression in the P and B samples (Figure 6c). The promoter regions of CHS1 (LOC112175474), CHS4 (LOC112175453), F3′H (LOC112178545), and DFR (LOC112173668) all contain MYB-binding sites. This implies that all of these genes may be regulated by MYB transcription factors.

### 2.7. Association of Metabolites and Transcriptomics Analysis

To explore the transcriptome–metabolome relationship in the rose samples, we analyzed the correlation between DEGs and DAMs. KEGG enrichment analysis revealed consistent enrichment of differential genes and metabolites in phenylpropanoid, flavonoid, and anthocyanin biosynthesis pathways, indicating a close linkage between metabolite accumulation and differential gene expression (Figure 7a). To perform K-means cluster analysis, FPKM values of the genes were centered and normalized (Appendix A). To explore the trend of the metabolite relative content across fractions, we standardized the mean relative content of differential metabolites in each group by applying Z-score normalization, followed by K-means clustering analysis. The results showed that Sub-class 3 metabolites had higher relative levels in the Y samples, Sub-class 9 metabolites had higher relative levels in the P samples, and Sub-class 1 metabolites had the highest relative levels in the B samples (Figure 7b).

We screened an anthocyanin synthesis-related gene, DFR (LOC112173668), in Sub-class 3 genes and three metabolites in Sub-class 9 metabolites—pelargonidin-3,5-O-diglucoside, pelargonidin-3-O-glucoside, and cyanidin-3-O-(6″-O-p-coumaroyl-2″-O-xylosyl) glucoside (pme1793, pme3392, and Lmjp002491). Six anthocyanin synthesis-related genes—CHS (LOC112175474, LOC112175453), PAL (LOC112179338), CHI (LOC112185875), F3′H (LOC112178545), ANS (LOC112179310), and UFGT (LOC112198391)—and 19 metabolites were screened in Sub-class 1 metabolites, as shown in the Appendix A. To delve deeper into the interplay between genes and metabolites during anthocyanin synthesis, we performed a correlation analysis on genes and metabolites detected in distinct subgroups. We calculated Pearson correlation coefficients and plotted them in a nine-quadrant diagram, focusing on genes with metabolite differences exhibiting high correlation (Pearson > 0.8) within each subgroup (Appendix A). Genes and metabolites in quadrants 3 and 7 exhibited consistent patterns of differential expression. Detailed analysis of the nine-quadrant map indicated that in the P/W group, the gene DFR (LOC112173668), and metabolites such as pelargonidin-3-O-glucoside and pelargonidin-3,5-O-diglucoside resided in quadrant 3 (Appendix A). In the B/W group, the genes CHS (LOC112175474, LOC112175453), CHI (LOC112185875), F3′H (LOC112178545), and UFGT (LOC112198391), and a variety of metabolites (Figure 7c and Appendix A) are also present in quadrant 3. The above results suggest that these metabolite changes may be positively regulated by these genes. Therefore, these enzymes and metabolites encoding DEGs may be associated with different color formations in roses.

## 3. Discussion

### 3.1. Anthocyanins and Carotenoids Influence Rose Red and Yellow Color Formation

Flower coloration is significant, particularly in ornamental plants such as roses. The coloration of rose petals is due to the accumulation of anthocyanins, a key metabolite that influences flower color [18]. The different pigment contents in the petals of four rose varieties were identified using commercially available standards. We found that the petals of the P and B varieties contained high amounts of anthocyanins, the petals of the Y variety contained the highest amounts of carotenoids, while those of the W, P, and B varieties had almost no carotenoids. Moreover, flavonoids were present in the petals of all four varieties. Upon analyzing anthocyanin composition, we observed a higher diversity of anthocyanin species in the petals of the P and B varieties, whereas the petals of the Y variety exhibited a lower diversity. In addition, most anthocyanins were upregulated in the petals of both the P and B varieties, suggesting that anthocyanins are important for the formation of pink and black-red colors (Appendix A). Prior investigations on rose petal coloration revealed that the combined content of cyanidin 3,5-O-diglucoside and peonidin 3,5-O-diglucoside determines the intensity of hues, ranging from pastel pinks and violets to lighter or deeper shades of red [19,20]. We analyzed DAAs in the W, P, and B varieties and found that most of the anthocyanins were accumulated in only the petals of the P and B varieties compared with the W variety. Cyanidin-3-O-(6″-O-p-coumaroyl-2″-O-xylosyl) glucoside and cyanidin-3-O-(6″-O-p-coumaroyl) glucoside were exclusively present in the petals of the W variety. Pelargonidin-3,5-O-diglucoside and pelargonidin-3-O-glucoside accumulated solely in the petals of the P variety, while the majority of anthocyanins, excluding these four, accumulated abundantly in the petals of the B variety (Figure 4b). In some species, anthocyanins are also present in white flowers. For instance, in Petunia, the white mutant possesses limited anthocyanin content [21]. Our analysis also showed the presence of small amounts of anthocyanins in white flowers, aligning with previously documented observations and demonstrating the complexity of pigment interactions in determining rose coloration.

Carotenoids are another abundant natural pigment. This pigment accumulates in the petals of the Y variety, resulting in their distinct coloration, as demonstrated in this study. Analysis via LC-MS revealed lutein, zeaxanthin, α-carotene, β-carotene, and ζ-carotene as the primary pigment components in the petals of the Y variety (Figure 2). The existence of a functional carotenoid biosynthesis pathway in the W and B rose samples is evident. However, the low carotenoid content in W-type roses precludes visible coloration, while the overwhelming anthocyanin levels in B-type roses mask carotenoid hues. Conversely, P-type roses boast high concentrations of both anthocyanins and carotenoids, indicating a collaborative role in determining rose coloration.

### 3.2. Structural Genes and Metabolites Regulating Flower Color in Roses

Anthocyanin and carotenoid biosynthesis is governed by structural genes within the biosynthetic pathways. Our results show that most anthocyanin biosynthetic genes, including CHS, CHI, F3′H, DFR, ANS, and UFGT, are highly expressed in P-type and B-type roses. Notably, DFR, ANS, and UFGT genes, which are crucial for anthocyanin synthesis and accumulation, are minimally expressed in white roses (Figure 4). DFR converts dihydroflavonol to leucoanthocyanidin, which is then modified to form pelargonidin, cyanidin, and delphinidin [22]. Studies have shown that in strawberries, the higher the expression of the *DFR* gene, the higher the anthocyanin content [23]. In safflower (*Carthamus tinctorius* L.), the expression of DFR is upregulated during the process of its flowers transitioning from yellow to red [24]. Anthocyanin synthase (ANS), a 2-oxoglutarate (2OG) iron-dependent oxygenase downstream of DFR, catalyzes the transformation of colorless anthocyanins into colored ones, marking the transition from achromatic to chromatic compounds. Plant ANS enzyme-coding sequences have been detected, and ANS expression correlates with anthocyanin accumulation [25,26,27]. In apples, decreased ANS expression results in reduced anthocyanin biosynthesis [28]. Research indicates that McANS plays a crucial role in petal pigmentation, suggesting the transcriptional activity of this gene significantly impacts the reddish hue of begonia blossoms [29]. In addition, UFGT is a glycosyltransferase that determines the site of glycosylation and is important for the stability and water solubility of phycocyanins [30]. In grapes, UFGT regulates anthocyanin synthesis during the reddening process [31]. Reduced expression of the UFGT gene leads to decreased anthocyanin content, as exemplified in tobacco, where low NtUFGT expression results in flowers with minimal anthocyanin content, appearing nearly white [32].

Utilizing K-means cluster analysis and nine-quadrant plots, we examined the correlation between anthocyanin synthesis genes and their corresponding metabolites. Notably, both differential genes and metabolites were enriched in biosynthetic pathways involved in phenylpropanoids, flavonoids, and anthocyanins (Figure 7a). A thorough examination of structural genes and metabolites involved in anthocyanin synthesis revealed that in the P/W group, upregulation of DFR expression led to increased levels of pelargonidin-3,5-O-diglucoside, pelargonidin-3-O-glucoside, and cyanidin-3-O-(6″-O-p-coumaroyl-2″-O-xylosyl) glucoside. Conversely, in the B/W group, upregulation of CHS, CHI, F3′H, and UFGT expressions resulted in the upregulation of multiple metabolite levels (Figure 7c and Appendix A). In conclusion, we speculate that an augmentation in the expression of structural genes in the anthocyanin biosynthesis pathway boosts the abundance of associated metabolites, ultimately shaping the coloration of rose blossoms.

To evaluate the putative influence of carotenoid biosynthesis pathway genes on rose coloration, we constructed a heatmap that portrays the expression patterns of crucial genes participating in carotenoid biosynthesis (Figure 5). Notably, PSY, a pivotal enzyme regulating the carotenoid synthesis pathway, emerged as a potential factor promoting carotenoid accumulation in the petals of the Y and B varieties [33]. PSY expression levels were upregulated during citrus fruit coloration [34]. ZISO and LYCB play a role in carotenoid synthesis in the petals of the Y variety (Figure 5). LC-MS analysis demonstrated that the concentrations of α-carotene, β-carotene, ζ-carotene, and zeaxanthin were notably elevated in the petals of the Y variety compared with the other three rose varieties (Figure 2). Overexpression of IbLCYB2 significantly increased α-carotene, β-carotene, lutein, and zeaxanthin content in sweet potato [35]. In summary, the increased expression of PSY, ZISO, and LYCB in the petals of the Y variety is likely to increase the carotenoid and zeaxanthin content, resulting in yellow hues. Cultivating roses with a variety of flower colors is an important breeding objective. The key to achieving this is to manipulate flower color by adjusting the expression of anthocyanin or carotenoid biosynthesis genes, thereby altering the levels of relevant metabolites.

### 3.3. Transcription Factors and Structural Genes Regulating Rose Flower Color

The MBW protein complex, encompassing MYB, bHLH, and WD40, serves as a conserved regulatory element in anthocyanin biosynthesis across higher plants. MYB and bHLH TFs, which are widely present in eukaryotes, belong to one of the most prominent families of plant transcriptional regulators [36,37]. As integral components, bHLHs interact with specific MYB proteins. Mutations in MYB-binding regions can trigger bHLH transcriptional activity, indicating a regulatory function of MYB [38]. MBW components have been discovered in diverse plant species, including Arabidopsis [39], banana [40], and grape [41]. In this study, 17 bHLH, 19 MYB, 23 NAC, and 11 WRKY transcription factors were identified as regulators of rose anthocyanin synthesis. MYB TFs play a crucial role in regulating anthocyanin production in the phenylpropanoid pathway. In anthocyanin biosynthesis, some MYB-TFs function as activators (R2R3-MYB), while others repress gene expression (R2R3-MYB and R3-MYB) [42]. For instance, PavMYB10.1 in sweet cherry (Prunus avium) activates anthocyanin biosynthesis by upregulating downstream regulators and structural genes [43]. Similarly, RcMYB1 overexpression boosts anthocyanin accumulation in white rose petals and tobacco leaves [44]. HaMYB1 regulates anthocyanin accumulation to deepen flower colour in sunflower [45]. MYB-TFs inhibiting anthocyanin synthesis include PhMYBx [46], peach PpMYB140 [47], PpMYB17-20 [48], and apple MdMYBPA1 [49]. Research has demonstrated that in peony, PsMYB44 exerts a negative regulatory effect on anthocyanin biosynthesis by directly binding to the PsDFR promoter, thereby inhibiting the formation of anthocyanin spots [50]. In this study, RhMYB1 and RhMYB113 exhibited higher expression in the petals of the P and B varieties, along with upregulated expression of structural genes in the anthocyanin synthesis pathway, including CHS, CHI, F3′H, DFR, ANS, and UFGT (Figure 4 and Figure 6). Moreover, promoter analysis revealed MYB-binding sites within the promoters of CHS, F3′H, and DFR, indicating potential regulation by MYB TFs (Figure 6c). Our results suggest that RhMYB1 and RhMYB113 regulate red pigmentation by upregulating genes essential for anthocyanin biosynthesis. Furthermore, overexpression of SlNAC1 boosts the expression of SlLCYb, a key gene in carotenoid synthesis, leading to enhanced β-carotene levels [51]. In this study, 17 of the 23 NACs were highly expressed in the petals of the Y variety, suggesting that NACs mainly participate in yellow color formation. In addition, the carotenoid synthesis pathway structural genes PSY, ZISO, and LYCB were highly expressed in the petals of the Y variety. We hypothesize that the NAC TFs may contribute to the development of rose yellow color by increasing the transcriptional levels of crucial genes such as PSY, ZISO, and LYCB.

## 4. Materials and Methods

### 4.1. Plant Materials

The petals of four Rosa hybrids with typical characteristics were selected for this study: Bai Xueshan (white), Jin Xiangyu (yellow), Litchi (pink), and Black Magic (black-red). These four rose varieties were planted in a greenhouse to ensure identical growth conditions. Three individual plants from each variety were selected, and their semi-open petals were carefully detached and collected. The gathered specimens were promptly flash-frozen using liquid nitrogen and kept in a −80 °C ultra-low-temperature freezer for subsequent RNA and pigment extraction.

### 4.2. Quantification of the Total Anthocyanin Amount

Utilizing the pH-differential spectrum technique, the total anthocyanin content in the gathered samples was assessed, as outlined by Rapisarda [52]. A 100 mg aliquot of frozen samples was pulverized and then individually processed with 2 mL of a pH 1 buffer (containing 0.15 M HCl and 0.05 M KCl) and 2 mL of a pH 4.5 buffer (composed of 0.24 M HCl and 0.4 M sodium acetate). These mixtures were centrifuged at 13,000× *g* for 5 min at 4 °C. Subsequently, the absorbance of the diluted supernatants was measured at 510 nm. The total anthocyanin concentration was calculated based on the formula: Amount (mg/g FW) = (ΔD × 484.82 × 1000)/24,825 × DF, where ΔD denoted the absorbance difference between the pH 1 and pH 4.5 solutions.

### 4.3. Extraction and Quantification of Carotenoids

Rose petal carotenoids were extracted and quantified using a revised protocol based on prior studies [53]. Lyophilized samples (1.0 g) of each rose variety were harvested for physiological analysis. To isolate total carotenoids, tissues were pulverized and dissolved in 10 mL of a 60/40 (*v*/*v*) n-hexane–acetone mixture. After centrifugation at 4000× *g* for 5 min, the clarified supernatant was transferred to a fresh opaque tube. The residue was re-extracted with fresh solvent until colorless. The carotenoid content was quantified at 450 nm using the following formula: (OD_450_)/0.25 (mg/mL).

To accurately assess the carotenoid concentration, lyophilized plant material was finely pulverized. Then, 50 mg of dried tissue powder was extracted in 20 mL of a solvent mixture (acetone, n-hexane, ethanol) with an internal standard. After 20 min of vortex mixing and centrifugation, the supernatants were collected. The supernatants were evaporated to dryness under nitrogen, redissolved in 2 mL of methanol–MTBE, and filtered through a 0.22 μm membrane for LC-MS analysis. The analysis of LC-MS was performed using an LC device (Shimadzu LC30AD, Waltham, MA, USA) and a mass spectrometer (TRIPLE QUAD 5500, Nakagyo-ku, Kyoto, Japan).

HPLC (RIGOL scientific, Beijing, China) analytical parameters were optimized as follows: employing a YMC C30 column (3 μm, 100 mm × 2.0 mm internal diameter) with a 2 μL injection volume. Gradient elution employed solvents A (acetonitrile with 0.01% BHT and 0.1% formic acid) and B (methyl tert-butyl ether with 0.01% BHT). The gradient ranged from 0% B (0–3 min) to 70% B (3–5 min), peaking at 95% B (5–9 min) and returning to 0% B (11–12 min). The column temperature was set at 28 °C, and the flow rate was maintained at 0.8 mL/min.

Utilizing the AB Sciex Instruments LC-MS/MS system, the carotenoid concentrations were detected by following previously described procedures (Xiong et al., 2019). The APCI source was set to APCI+ with a 350 °C source temperature. The curtain gas flow (CUR) was calibrated to 25.0 psi, while the DP and CE settings for individual MRM transitions were fine-tuned. Each elution cycle was tailored to monitor specific MRM transitions, matching the carotenoids eluted during that phase.

### 4.4. Identification and Quantitative Analysis of Metabolites

The multiple response monitoring (MRM) mode was used for the quantitative analysis of metabolites. Initially, we procured metabolic mass spectrometry data for various samples. Subsequently, we integrated peak areas across all mass spectrometry peaks and processed the integration and correction of identical metabolite peaks in different samples [54]. Utilizing analyst 1.6.3 software, we analyzed the processed mass spectrometry data to qualitatively and quantitatively assess metabolites within the samples, leveraging local metabolic databases. Metabolites exhibiting significant regulation were screened based on VIP ≥ 1 and absolute Log2FC (fold change) ≥ 1, with VIP values derived from OPLS-DA analysis. Furthermore, the OPLS-DA results encompassed score plots and permutation plots generated by the R package MetaboAnalystR. Prior to data analysis, a log2 transformation and mean centering were performed, and a permutation test (totaling 200 permutations) was executed to mitigate overfitting.

### 4.5. RNA Extraction and Quantitative Real-Time PCR Analysis

To extract the total RNA from rose petal samples, trizol reagent (Takara, Dalian, China) was utilized, adhering to protocols established in prior research [55]. DNA contamination was removed using DNase I (Invitrogen, Carlsbad, CA, USA). Synthesis of first-strand cDNA was achieved from 1 μg of RNA by employing M-MLV reverse transcriptase (Promega, Beijing, China) and oligo(dT)20 primers. Real-time quantitative PCR was then performed using the SYBR Premix Ex Taq kit (Takara, Dalian, China) on the CFX96™ Real-Time System (Bio-Rad, Hercules, CA, USA). RhUBI6 served as the internal reference gene [56], and relative expression levels were determined via the 2^−ΔΔCT^ method [57]. Primer sequences are detailed in Appendix A.

### 4.6. RNA-Sequencing and Data Analysis

For this study, four distinctive rose cultivars were chosen, and petal samples were gathered for RNA sequencing (RNA-seq) analysis. To ensure credibility, three biological replicates were collected per cultivar for transcriptome analysis. These samples were then sent to Wuhan MetWare Biotechnology Co., Ltd. (Wuhan, China) for RNA sequencing. Gene model annotation files were derived by comparing the assembled genes with public databases, including Nr, Swiss-Prot, KOG/COG, KEGG, and GO [58,59]. Expression levels were evaluated using FPKM [60], and differentially expressed genes were identified based on a 2-fold change criterion. Plant transcription factor prediction was carried out utilizing iTAK 1.7a software.

### 4.7. Statistical Analysis

All data were presented as the mean ± SD (error bars indicated the standard deviations of the means) from at least three biological replicates. This study used GraphPad Prism 9.0 (GraphPad Software, https://www.graphpad.com/) for statistical analysis. One-way analysis of variance (ANOVA) was performed by Duncan’s tests, with significance set at *p* < 0.01.

## 5. Conclusions

Our study reveals the formation of different rose petal colors through the integration of metabolomics and transcriptomics data. Variations in anthocyanin accumulation contribute to red coloration, while carotenoids play a key role in yellow coloration. A total of 22 major anthocyanins were identified, with distinct expression patterns in W, P, and B petal varieties. Notably, pelargonidin derivatives are primary contributors to pink hues, while specific cyanidin and peonidin derivatives accumulate exclusively in B petals, serving as crucial components for red color development. Additionally, lutein, zeaxanthin, and other carotenoids contribute significantly to yellow coloration. Through analysis of the anthocyanin and carotenoid synthesis pathways, we identified 11 and 12 structural genes, respectively, along with differentially expressed transcription factors (TFs), including MYB, WRKY, NAC, and bHLH. We constructed a model for rose color regulation (Figure 8), proposing that NAC TFs regulate carotenoid biosynthesis genes, facilitating lutein and zeaxanthin accumulation for yellow petals, while MYB TFs up-regulate anthocyanin synthesis genes, leading to the formation of red petals. These findings elucidate the molecular mechanisms and regulatory networks underlying rose anthocyanin and carotenoid biosynthesis, providing a foundation for future breeding of novel rose varieties.

## Figures and Tables

**Figure 1 plants-13-03353-f001:**
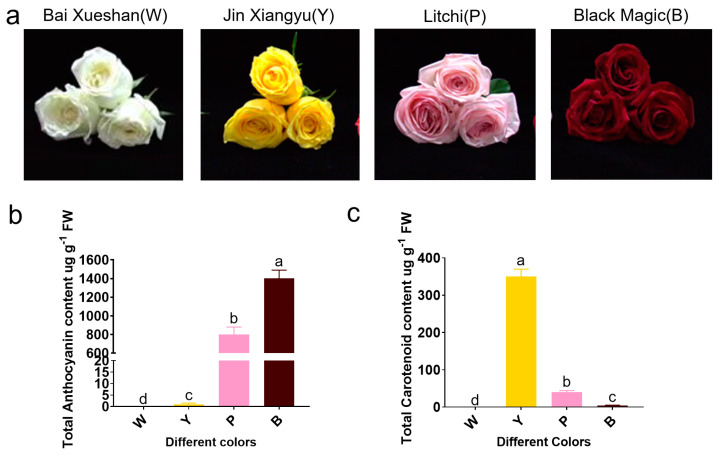
Characteristics of representative (*Rosa hybrid.*) flowers used in this study. (**a**) Petal color of four rose varieties. The white variety (Bai Xueshan), the yellow variety (Jin Xiangyu), the pink variety (Litchi), and the black variety (Black Magic). (**b**,**c**) The contents of anthocyanins and carotenoids were quantitatively determined by spectrophotometry. The error bar indicates the standard error of the mean (*n* = 3). ANOVA was used to assess differences between samples, and bars with different letters represent significant differences, *p* < 0.01.

**Figure 2 plants-13-03353-f002:**
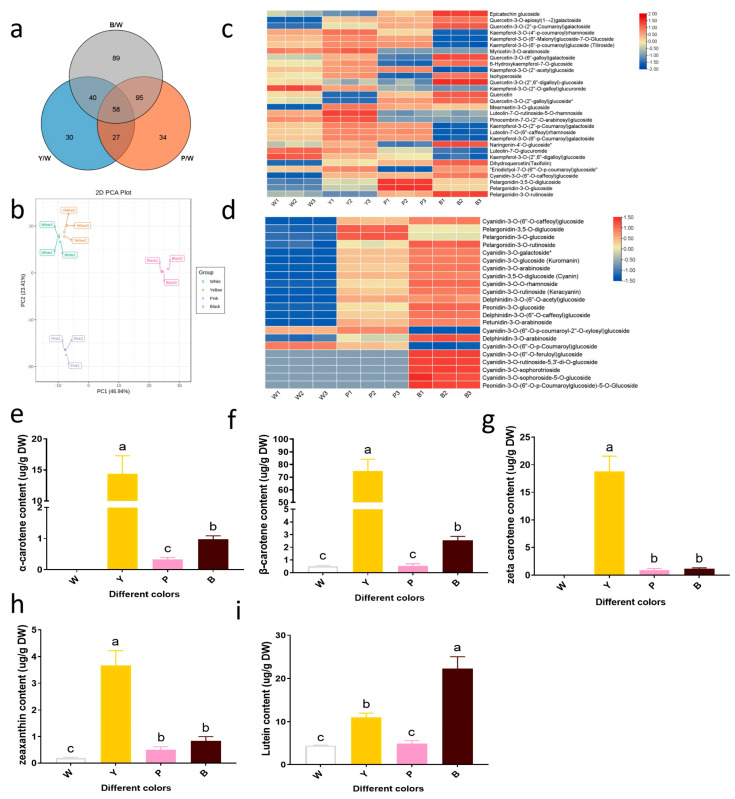
Analysis of differentially accumulated metabolites within the metabolome. (**a**) Venn analysis of differential metabolites. (**b**) Principal component analysis of metabolites. (**c**) Heat map analysis of differentially accumulated flavonoids by TBtools. (**d**) Heat map analysis of differentially accumulated anthocyanins by TBtools. (**e**–**g**) Contents of different carotene. (**h**) Zeaxanthin content. (**i**) Content of lutein. The error bar indicates the standard error of the mean (*n* = 3). ANOVA was used to assess differences between samples, and bars with different letters represent significant differences, *p* < 0.01.

**Figure 3 plants-13-03353-f003:**
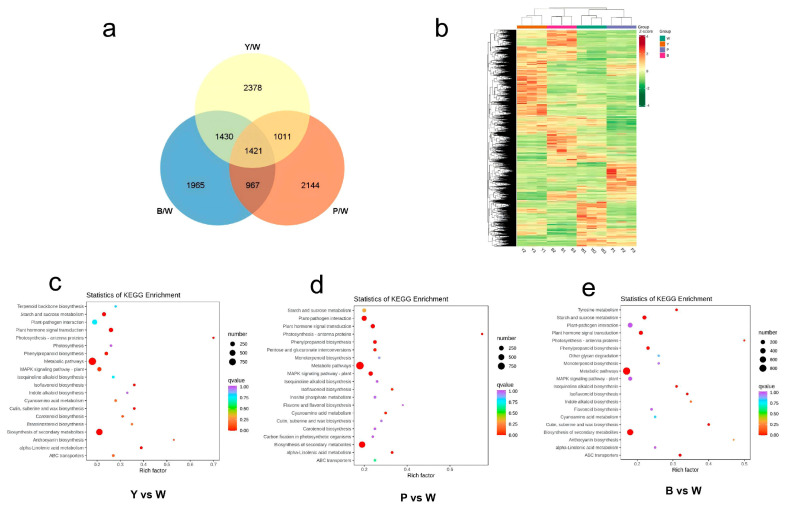
Analysis of gene expression profiles of four rose varieties. (**a**) Venn diagram of differentially expressed transcripts between four varieties. (**b**) Hierarchical cluster analysis was conducted to assess the DEGs across petals of various colors. The Log2 FPKM values are represented using a color scale ranging from green (−1.5, indicating inhibition) to red (1.5, indicating induction). The presented data represents the mean of three biological replicates. (**c**–**e**) For the comparison groups of (Y vs. W), (P vs. W), and (B vs. W), KEGG enrichment analysis was performed. The “GeneRatio” metric signifies the proportion of DEGs related to the KEGG pathway compared to the overall count of DEGs.

**Figure 4 plants-13-03353-f004:**
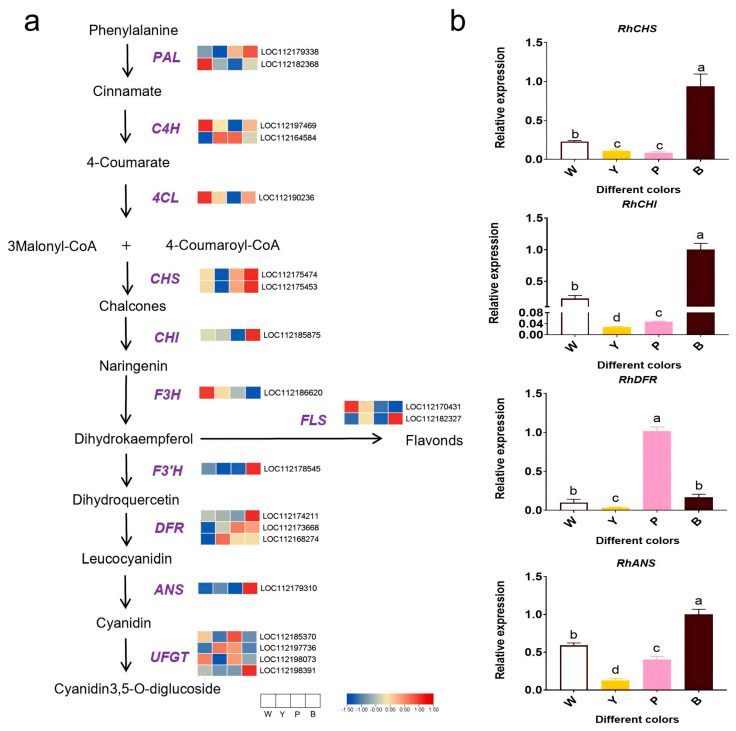
Differential expression of anthocyanin biosynthesis pathway genes in rose petals. (**a**) Biosynthetic pathways of rose anthocyanins and heat maps of genes involved in anthocyanin biosynthesis. The histogram shows the level of gene expression (FPKM value). The color keys represent a single gene expression value normalized to a Z-score (between −1.5 and 1.5). The blue and red charts show the low and highest expressed structural genes, respectively. The data are the average of the three biological replicates. (**b**) qRT-PCR detection of anthocyanin synthesis gene relative expression analysis. The error bar indicates the standard error of the mean (*n* = 3). ANOVA was used to assess differences between samples, and bars with different letters represent significant differences, *p* < 0.01.

**Figure 5 plants-13-03353-f005:**
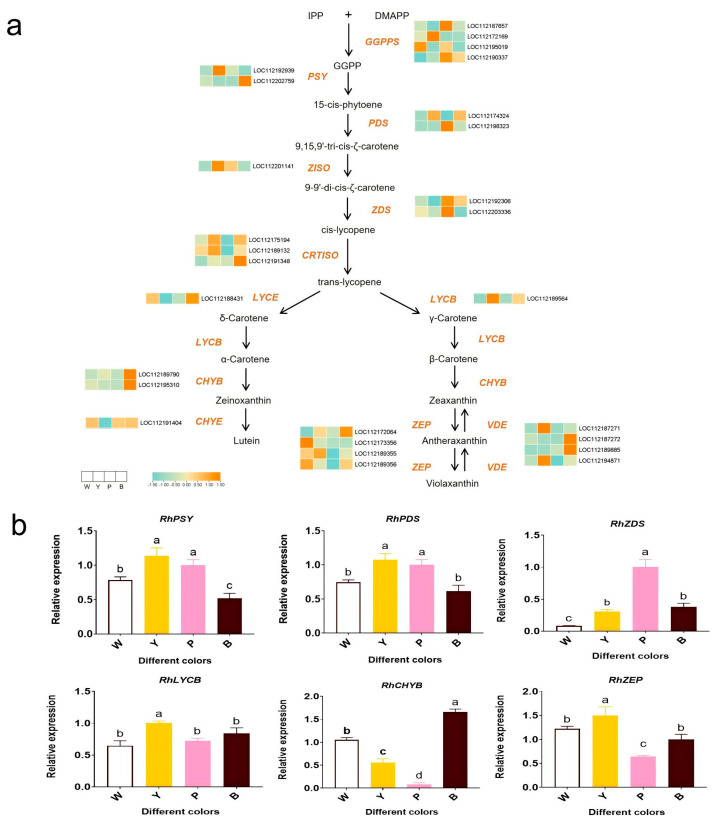
Discovery of genes involved in carotenoid biosynthesis in rose petals. (**a**) Heat map of the carotenoid bioanabolic pathway and transcription levels of carotenoid biosynthesis genes in rose. The histogram shows the FPKM value. The color keys represent a single gene expression value normalized to a Z-score (between −1.5 and 1.5). The green and orange charts show the low and highest expressed structural genes, respectively. Three experiments were replicated in our study. (**b**) Analysis of relative expression levels of differentially expressed genes related to carotenoid biosynthesis by qRT-PCR. The error bar indicates the standard error of the mean (*n* = 3). ANOVA was used to assess differences between samples, and bars with different letters represent significant differences, *p* < 0.01.

**Figure 6 plants-13-03353-f006:**
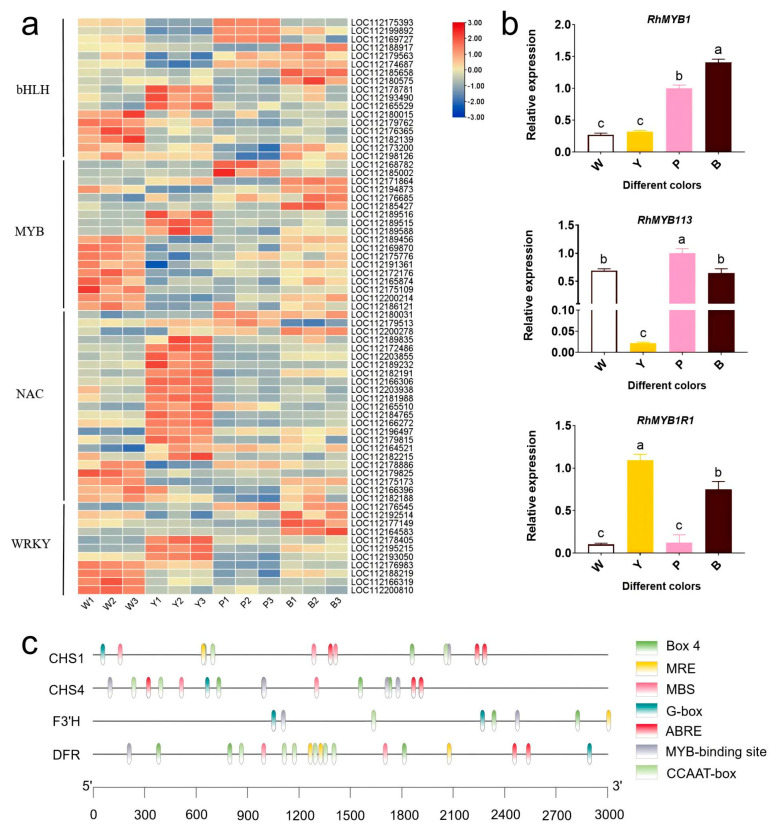
Rose flower color regulates transcription factors. (**a**) Varying expression of transcription factors linked to flavonoid metabolic pathways. (**b**) The relative expression levels of transcription factors related to flavonoid metabolic pathways were analyzed by qRT-PCR. (**c**) Analysis of cis-acting elements in promoters of structural genes of the anthocyanin synthesis pathway. The error bar indicates the standard error of the mean (*n* = 3). ANOVA was used to assess differences between samples, and bars with different letters represent significant differences, *p* < 0.01.

**Figure 7 plants-13-03353-f007:**
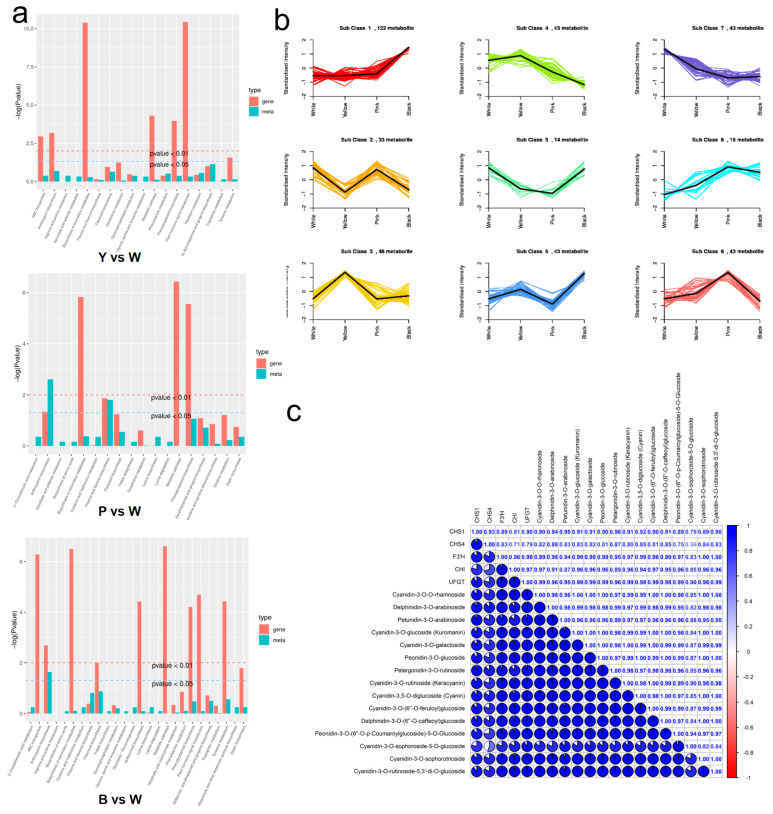
Associations between metabolites and transcriptomic analysis. (**a**) KEGG enrichment pvalue histogram. (**b**) Differential metabolite Kmeans map. (**c**) Analysis of anthocyanin content and key gene expression correlation in anthocyanin synthesis.

**Figure 8 plants-13-03353-f008:**
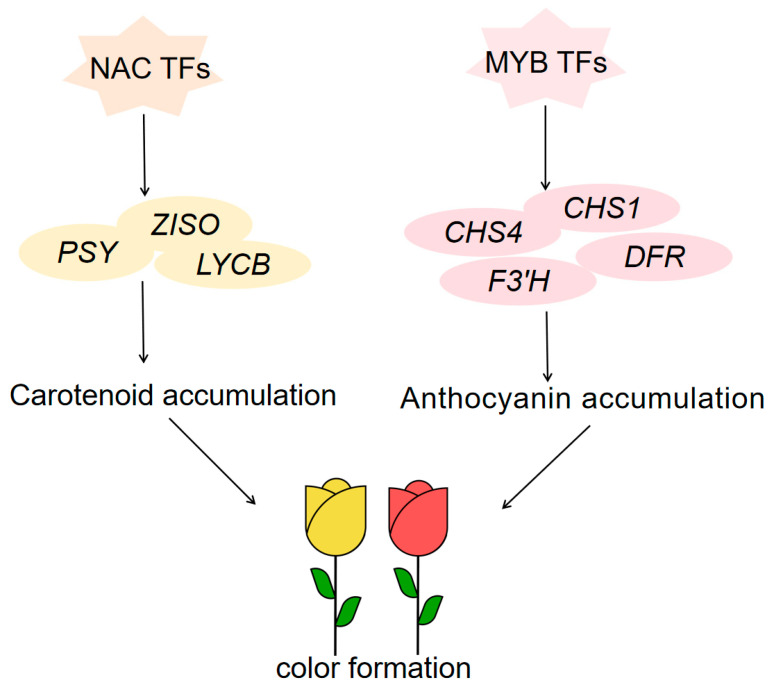
Rose flower color regulation model.

## Data Availability

Data will be made available on request.

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
