# Peer review of "Novel Insights into Pigment Composition and Molecular Mechanisms Governing Flower Coloration in Rose Cultivars Exhibiting Diverse Petal Hues"

_plants, 2024, doi:10.3390/plants13233353_

Round 1

Reviewer 1 Report

Comments and Suggestions for Authors

Title: Novel Insights into Pigment Composition and Molecular Mechanisms Governing Flower Coloration in Rose (Rosa hybrid.) Cultivars Exhibiting Diverse Petal Hues: please delete the scientific name and use it only in keywords.

Abstract: not easily readible – too much acronyms.

Introduction: it is generally caotic and each paragraph seems separated (completely) from another. There are several acronims never specified. Please specify them and try to make the introduction a good point to invite the reader to understand as bette ras possible the importance of your study.

Results: figures are not readible and I can’t evaluate results section. Please resubmit the manuscript by using high quality images. In the text it is important to highlight the results by indicating differences among thesis supported by percentage (-XX, +XX%... ) and when you have results and discussion you should indicate it in the head of the paragraph (otherwise you should add the “discussion” section). I suggest to separate each cultivar by a comma or other symbols different from “/”. In general your results are very complex, not ordered, not easy to read and mixed sometimes with a sort of discussion (if so, it is certainly a too short discussion!) and material and methods. Additionally, It is very difficult to check the quality of the paragraph without good figures and obtained data well described.

Material and methods

Plant materials: not sufficiently described. Please add the variety, age of the plant, soil details, environmental conditions. The number of plants used.

Milliliters>> mL.

Why some graphs are presented as FW and other in DW

In general all methods presented (included HPLC)  missed all details about instruments used and well detailed informations.

Invert 4.5 and 4.6

Did you verify normality of data?

DISCUSSION: please support better your results by using appropriated references

Comments on the Quality of English Language

Authours must check the english language (supported by native english speak!!) because the text is really difficult to read and is not well understandable.

Reviewer 2 Report

Comments and Suggestions for Authors

This study examines the pigmentation of rose petals, finding that pink and black-red varieties contain high levels of anthocyanins, while white and yellow petals have lower concentrations. Twenty-two key anthocyanin components, such as Cyanidin and Pelargonidin, were primarily identified in the pink and black-red petals. Five carotenoid species, notably zeaxanthin, were enriched in yellow petals. They further showed that RNA-seq revealed that the MYB transcription factor positively regulates anthocyanin biosynthesis genes, while the NAC transcription factor enhances carotenoid biosynthesis. These findings illuminate the molecular mechanisms of rose pigmentation and suggest potential pathways for rose breeding. However, addressing the above comments and questions will enhance the clarity and robustness of the research, making it more accessible to a broader audience and improving its impact within the field.

1. On page 3: 2. What is this: Results This section may be divided by subheadings. It should provide a concise and precise description of the experimental results, their interpretation, as well as the experimental conclusions that can be drawn.

2. What specific statistical thresholds were used to determine significance in the KEGG enrichment analysis?

3. The differentiation between Sub Class 3, Sub Class 9, and Sub Class 1 metabolites is an important aspect. Can you provide a brief description of what each subclass represents? How were they categorized?

4. How were the DFR and other specific genes and metabolites selected for discussion? Were they chosen based on statistical significance, biological relevance, or another criterion?

5. The data is solely computational, and it lacks experimental validations such as gene knockout or overexpression experiments to assess the functional roles of key DEGs (e.g., DFR, CHS) in anthocyanin production.

6.  In figure 8, the auhtors tried to show the regulatory model of floral color, but its very simple and no proper mechanistic insights are provided. It is suggested that authors should do more work on developing a more logical model.

7. The title of subsection 2.2. should be corrected. 

8. The quality of Figure 2 is really bad. It needs to simplified and make it clearer.

9. Where are the HPLC peaks?

10. In the method section, the analysis of gene expression, the criteria and details of statistical methods are absent. 

11. The conclusions should be revised. The existing one is too long, and it does not mention any future perspectives of this study. 

Comments on the Quality of English Language

Minor corrections with sentence structure are required.

Round 2

Reviewer 1 Report

Comments and Suggestions for Authors

Dear authors

Thanks to revise your manuscript. It is very difficult to understand what you have changed and what is going to remain in the text. Anyway, here few comments.

RESULTS

Be careful to separate correctly each word (i.e. L120 and similar).

I still have concern regarding the figures. I’m not conviced that the resolution is the correct one (especially the fig. 2 and 3).

MATERIAL AND METHODS

Please use the measure units abbreviations. Gramà g

Paragraph 4.7. Please, descrive the statistical analysis accurately.

Througout the text I saw grammar errror, missed spaces and similar, but the manner in which you revised the ms did not allow a proper identification of potential mistakes.

Comments on the Quality of English Language

The text require a native speaker revision

Reviewer 2 Report

Comments and Suggestions for Authors

The manuscript is significantly improved and it can be considered for publication with incorporating the following minor changes. 

1. The authors needs to add some discussion about the flower color transition and changes in flavonoid oathway gene expression in other flowering plants for comparision. For example, the following reference would be greatly helpful: https://doi.org/10.3390/ijms252211903

2. The manuscript still suffer some grammatical mistakes. 

3. I aslo suggest to have some recent literature added in the discussion section.
